# Environment-specificity and universality of the microbial growth law

Qirun Wang[1] & Jie Lin [1,2✉]

As the nutrient quality changes, the fractions of ribosomal proteins in the proteome are usually positively correlated with the growth rates due to the auto-catalytic nature of ribosomes. While this growth law is observed across multiple organisms, the relation between the ribosome fraction and growth rate is often more complex than linear, beyond models assuming a constant translation speed. Here, we propose a general framework of protein synthesis considering heterogeneous translation speeds and protein degradations. We demonstrate that the growth law curves are generally environment-specific, e.g., depending on the correlation between the translation speeds and ribosome allocations among proteins. Our predictions of ribosome fractions agree quantitatively with data of *Saccharomyces cerevisiae*. Interestingly, we find that the growth law curve of *Escherichia coli* nevertheless appears universal, which we prove must exhibit an upward bending in slow-growth conditions, in agreement with experiments. Our work provides insights on the connection between the heterogeneity among genes and the environment-specificity of cell behaviors.

[1] Center for Quantitative Biology, Academy for Advanced Interdisciplinary Studies, Peking University, Beijing 100871, China. [2] Peking-Tsinghua Center for Life Sciences, Academy for Advanced Interdisciplinary Studies, Peking University, Beijing 100871, China. ✉email: linjie@pku.edu.cn

Cells can adapt to different environments and alter the expression levels of multiple genes. The genome-wide gene expression profile can change dramatically as cells switch between different environments. However, proliferating cells, including bacteria and unicellular eukaryotes, exhibit a growth law as the nutrient quality changes: the fraction of ribosomal proteins in the proteome ($\phi_R$) and the growth rate ($\mu$) are positively correlated. The growth law curve ($\phi_R$ vs. $\mu$) is often fit by a linear relation, $\phi_R = \mu/\kappa + \phi_0$[1-6], which can be rationalized by a simple translation model (STM): ribosomes are engaged in translation with a constant translation speed that is proportional to $\kappa$[2,4]. $\phi_0$ represents the fraction of inactive ribosomes that are not producing proteins, independent of environments in the STM. While the STM is simple and intuitive, it does not always provide a good empirical fitting to the experimental growth law curves, e.g., it apparently breaks down in slow-growth conditions of Escherichia coli (doubling time longer than 60 min at 37 °C) in which more ribosomes are produced than the prediction of the STM[7]. A similar breakdown was also observed for other bacteria[1].

We remark that two important biological features beyond the STM are crucial to interpreting the growth law curve, as we show in this work. The first is the heterogeneous translation speeds of ribosomes producing different proteins. Recent studies showed that the translation speeds are highly heterogeneous among different proteins due to multiple mechanisms, including codon usages[8] and amino acid compositions[9]. In particular, the translation speeds of ribosomal proteins are much slower than the average translation speed over non-ribosomal proteins due to the abundance of positively charged amino acids in ribosomal proteins[9]. Nowadays, the ribosome profiling technique allows us to quantify the allocation of ribosomes toward the production of different proteins. These experimental techniques enable us to rethink the growth law in the presence of heterogeneity in translation speeds[9].

The second feature is finite protein degradation rates. The STM neglects protein degradation and predicts that at zero growth rate, $\phi_R = \phi_0$ so that all ribosomes are inactive. However, this contradicts experiments of nongrowing bacteria in which translation activities are observed[10]. Protein degradation must be considered at zero growth rate to balance protein production to ensure a constant protein mass. Therefore, protein degradation must be important to the growth law, at least in slow-growth conditions.

In this work, we show that the heterogeneous translation speeds and protein degradations significantly influence the growth law by introducing a general theoretical framework of protein synthesis. We find that the fraction of ribosomal proteins $\phi_R$ depends not only on the growth rate but also on the statistical properties of environments. Besides the growth rate, $\phi_R$ depends on two correlation coefficients among proteins. One is between the translation speeds and ribosome allocations towards the production of different proteins. The other is between the degradation rates and mass fractions of proteins. Both correlation coefficients are environment-specific. We compute the above correlation coefficients using proteomics and ribosomal profiling datasets of S. cerevisiae[11]. Interestingly, we find that the correlation between the translation speed and ribosome allocations becomes stronger when the growth rate decreases; cells tend to produce more proteins with higher translation speeds in poor nutrients. In contrast, the correlation between the protein degradation rates and mass fractions is almost independent of growth rates.

We derive the general form of growth law involving the above correlations and demonstrate that for environments with similar correlation coefficients, the growth law curve is universal and has the following form, $\phi_R = (\mu + c_1)/(c_2\mu + c_3)$ where $c_1$, $c_2$, and $c_3$ are constants depending on the above correlation coefficients. In particular, $c_2$, which sets the nonlinearity of the growth law curve, is finite due to the slow translation speed of ribosomal proteins. We prove that a universal growth law curve must be monotonically increasing and convex. Surprisingly, we find that a universal growth law applies to E. coli and our theories justify the upward bending of the growth law curve of E. coli in slow-growth conditions relative to a linear line[7]. However, if the experiments are implemented in multiple environments with dramatically different correlation coefficients, the growth law curve is non-universal and environment-specific. Our analysis of experimental data suggests that this scenario may apply to S. cerevisiae. We fit the experimentally measured growth law curves by our model predictions, from which we can estimate the translation speed of ribosomal and non-ribosomal proteins. Consistent with direct experimental measurements[9], the estimated translation speed of ribosomal proteins is indeed much slower than non-ribosomal proteins.

## Results

**Model of protein synthesis.** Given a constant environment, we consider a population of cells with a constant growth rate, and the protein synthesis processes are in a steady state. Ribosome profiling allows us to quantify the fraction of ribosomes in the pool of total active ribosomes producing protein $i$, which we call ribosome allocation $\chi_i$. Here the index $i$ represents one particular protein $i$. Mass spectrometry also allows us to measure the mass fractions $\phi_i$ of all proteins in the proteome[12]. The translation speed of ribosomes on the corresponding mRNAs is $k_i$, which is the averaged mass of translated amino acids per unit time. Note that $k_i$ is averaged over the sequence of the corresponding mRNA so that each protein has one $k_i$. We also assume that protein $i$ degrades with a constant rate $\alpha_i$. The mass production rate of protein $i$ becomes

$$\frac{dM_i}{dt} = k_i\chi_i(R - R_0) - \alpha_i M_i. \tag{1}$$

Here $R$ is the number of ribosomes, and $R_0$ is the number of inactive ribosomes. Our model is summarized in Fig. 1.

In this work, we focus on the effects of heterogeneous translation speeds $k_i$ and finite degradation rates $\alpha_i$. Therefore, for simplicity, we assume them to be invariant of environments. We also mainly consider the effects of nutrient quality and do not consider the impact of antibiotics in this work, which can decrease the overall effective translation speed and increase $\phi_R$ as the growth rate decreases[4].

We define the total protein mass $M = \sum_i M_i$, and the protein mass fraction $\phi_i = M_i/M$. Using Eq. (1), we find the fraction of ribosomal proteins in the proteome in the steady state, (see detailed derivations in Methods)

$$\phi_R = \frac{m_R(\mu + \sum_i \alpha_i\phi_i)}{\sum_i k_i\chi_i} + \phi_0. \tag{2}$$

Here $\mu$ is the growth rate of the total protein mass $\mu = \dot{M}/M$, and $m_R$ is the total amino acid mass of a single ribosome. $\phi_0$ is the mass fraction of inactive ribosomes, which we assume to be constant for simplicity. In this work, $i = 1$ is reserved for ribosomal proteins so that $\phi_1 = \phi_R$, $k_1 = k_R$, and $\alpha_1 = \alpha_R$. Here, $k_R$ and $\alpha_R$ are the effective translation speed and degradation rate of the coarse-grained ribosomal protein averaged over all ribosomal proteins. They are approximately independent of environments due to the tight regulation of relative doses of different ribosomal proteins[13] and their generally low degradation rates. It is easy to find that if all proteins have the same translation speed ($k_i = k$ for all $i$) and protein degradations are negligible ($\alpha_i = 0$), Eq. (2) is reduced to the STM.

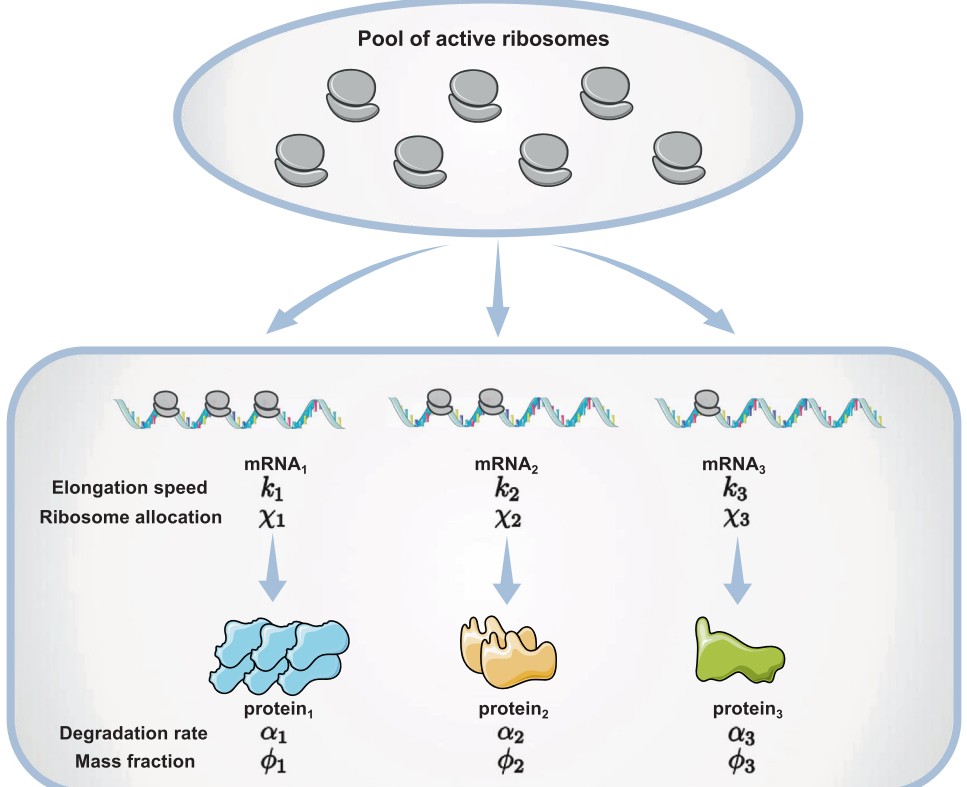

**Fig. 1 A schematic of the model.** Given a constant environment, cells allocate different fractions of active ribosomes ($\chi_i$) to translate mRNAs corresponding to different proteins. In general, the translation speeds $k_i$ are heterogeneous among proteins. $\alpha_i$ is the degradation rate of protein $i$. $\chi_i$, $k_i$ and $\alpha_i$ together determine the mass fraction of protein $i$. The ribosome allocation strategies reflect the adaptation of cells to different environments. In this schematic, we show three proteins for simplicity.

**Universal and non-universal growth law curves**. To disentangle the effects of heterogeneous translation speeds and protein degradations, we first simplify the model by taking $\alpha_i = 0$ for all proteins and only consider the effects of heterogeneous translation speeds $k_i$. We rewrite $\sum_i k_i \chi_i = k_R \chi_R + (1 - \chi_R) \sum_{i=2}^{N} k_i \widetilde{\chi}_i$ in Eq. (2). Here, $N$ is the number of proteins and $\chi_i = (1 - \chi_R) \widetilde{\chi}_i$ so that $\sum_{i=2}^{N} \widetilde{\chi}_i = 1$. In the following, we define $\langle k \rangle_\chi = \sum_{i=2}^{N} k_i \widetilde{\chi}_i$ as the $\chi$-weighted average translation speed over all non-ribosomal proteins. As we derive in Methods, the fraction of ribosomal proteins can be written exactly as a Hill function of the growth rate:

$$\phi_R = \frac{\mu}{a\mu + b} + \phi_0, \qquad (3)$$

where the expressions of $a$, $b$ are shown in Methods. We are particularly interested in the sign of $a$ because it determines the shape of the $\phi_R(\mu)$ curve. Interestingly, we find that $a \propto k_R - \langle k \rangle_\chi$. If $k_R$ is smaller than $\langle k \rangle_\chi$, $a$ is negative so that the second derivative of the $\phi_R(\mu)$ curve is positive. In other words, the $\phi_R(\mu)$ curve is upward bent in slow-growth conditions relative to a linear line.

$\langle k \rangle_\chi$ depends on both the elongation speeds $k_i$ and the ribosome allocations $\chi_i$. To find its value, we further rewrite $\langle k \rangle_\chi = \langle k \rangle (1 + I_{\chi,k})$. Here $\langle k \rangle$ is the arithmetic average of translation speeds over all non-ribosomal proteins. $I_{\chi,k}$ is a metric we use to quantify the correlation between the ribosome allocations and the translation speeds:

$$I_{\chi,k} = \frac{\langle \widetilde{\chi}_i k_i \rangle - \langle \widetilde{\chi}_i \rangle \langle k \rangle}{\langle \widetilde{\chi}_i \rangle \langle k \rangle}. \qquad (4)$$

Here, the bracket represents an average over all non-ribosomal proteins. Biologically, the higher $I_{\chi,k}$ is, the more ribosomes are allocated to mRNAs with higher translation speeds. Because the ribosomal allocations $\chi_i$ are generally different in different environments, we use $I_{\chi,k}$ to characterize an environment. Imagine that we grow cells in multiple environments with equal $I_{\chi,k}$. We find that as long as $I_{\chi,k}$ is not too close to $-1$, which we confirm later using experimental data, $a$ is always negative since the translation speed of ribosomal proteins $k_R$ is much lower than $\langle k \rangle$[9]. Therefore, Eq. (3) predicts an upward bending of the $\phi_R(\mu)$ curve in slow-growth conditions.

We verify the above theoretical predictions by numerically simulating the model of protein synthesis (Methods). The translation speeds are randomly sampled among proteins and fixed for all environments, with $k_R < \langle k \rangle$. We randomly sample $\chi_i$ for each environment and compute the resulting growth rate $\mu$ and protein mass fractions $\phi_i$. We show the results from environments with preselected $I_{\chi,k}$, which agree well with the theoretical formula Eq. (3) (Fig. 2a).

We also consider another simplified model in which the translation speeds are homogeneous, but protein degradation rates are finite and heterogeneous. We find that in this model, the growth law curve is linear with a reduced slope and increased intercept compared with the STM (see details in Methods). The actual shape of the growth law curve depends on the parameter $I_{\phi,\alpha}$, which is a metric to characterize an environment by quantifying the correlation between the protein mass fractions and degradation rates:

$$I_{\phi,\alpha} = \frac{\langle \widetilde{\phi}_i \alpha_i \rangle - \langle \widetilde{\phi}_i \rangle \langle \alpha \rangle}{\langle \widetilde{\phi}_i \rangle \langle \alpha \rangle}. \qquad (5)$$

Here, the bracket represents an average over all non-ribosomal proteins and $\widetilde{\phi}_i = (1 - \phi_R)\phi_i$. Biologically, a high $I_{\phi,\alpha}$ value means

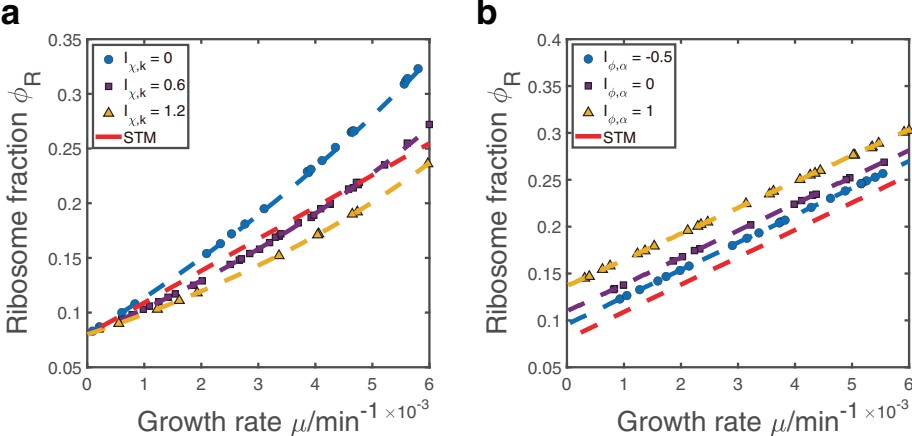

**Fig. 2 Numerical simulations of the growth law curves. a** We simulate the case of heterogeneous translation speeds without protein degradations and compare our numerical simulations with model predictions (dashed lines). Each data point has its own randomly sampled $\chi_i$, and we show the results with preselected $I_{\chi,k}$ values. The red dash line represents the predictions of the STM in which all proteins have the same translation speed $\langle k \rangle$. **b** Same analysis in which we simulate the case of finite protein degradation rates without heterogeneity in translation speeds.

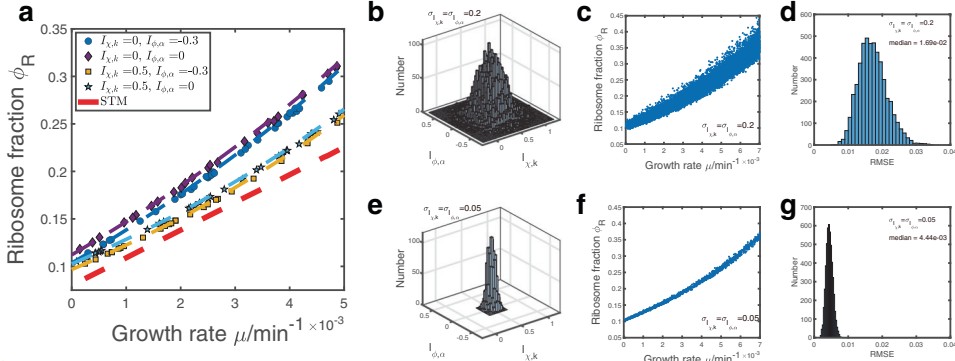

**Fig. 3 Numerical simulations of the growth law curves with both heterogeneous translation speeds and protein degradation rates. a** Numerical simulations with preselected $I_{\chi,k}$ and $I_{\phi,\alpha}$. The red dashed line is the prediction of the STM, and other dashed lines represent our model predictions. **b**, **e** Two-dimensional Gaussian distribution of randomly sampled $I_{\chi,k}$ and $I_{\phi,\alpha}$. The mean of $I_{\chi,k}$ is 0.5, and the mean of $I_{\phi,\alpha}$ is 0. The standard deviations $\sigma$ are indicated in the legends. **c**, **f** The resulting growth law curve where each point has randomly sampled $I_{\chi,k}$ and $I_{\phi,\alpha}$ from (**b**) and (**e**). **d**, **g** The distribution of the fitting RMSE corresponding to randomly chosen points in (**c**) and (**f**).

that the proteome are enriched with proteins with high degradation rates. We verify the above theoretical predictions by numerical simulations and randomly sample the protein degradation rates that are fixed for all environments. We show the results from environments with preselected $I_{\phi,\alpha}$ and our theoretical predictions Eq. (18) are nicely confirmed (Fig. 2b).

Finally, we turn to the full model with both the heterogeneities in the translation speeds and protein degradation rates. We find that the growth law curve has the following general form,

$$\phi_R = \frac{\mu + c_1}{c_2\mu + c_3},\qquad(6)$$

where the expressions of the constants, $c_1$, $c_2$ and $c_3$ are shown in Methods. We prove that given fixed $I_{\chi,k}$ and $I_{\phi,\alpha}$ (as long as they are not too close to −1), the growth law curve must be monotonically increasing and convex, which suggests an upward bending in slow-growth conditions (Methods). In particular, $c_2 \propto k_R - \langle k \rangle_\chi$, which means that it is the slower translation speed of ribosomal proteins than other proteins that generates the nonlinear shape of the growth law curve. The simulation results match the theoretical predictions (Fig. 3a). We note that the uncertainness of real environments often leads to random production of proteins and random allocation of ribosomes. To

address this question, we also simulate models in which noises exist in the translation speeds $k_i$ and the allocation fractions $\chi_i$. We find that both noises do not affect our conclusions qualitatively (Fig. S1). Note that adding noises to the translation speeds and allocation fractions only makes the resulting growth law curves even noisier and therefore does not affect our main conclusion that the growth law curve is generally environment-specific, as we show later.

In real situations, we remark that the actual growth curve shape depends on the particular environments. To verify this, we compute the resulting growth law curve with multiple environments, and the $I_{\chi,k}$ and $I_{\phi,\alpha}$ of each environment are randomly sampled from Gaussian distributions (Fig. 3b, e) (Methods). We find that when the Gaussian distributions have large standard deviations, the growth law curve is non-universal and depends on the particular chosen environments (Fig. 3c). This means that if we randomly pick some environments from Fig. 3c, the resulting growth law curves are generally different. In contrast, when the Gaussian distributions have small standard deviations, the growth law curve is well captured by our theoretical predictions Eq. (6) because the environments share similar $I_{\chi,k}$ and $I_{\phi,\alpha}$ (Fig. 3f). To quantify the effects of heterogeneous $I_{\chi,k}$ and $I_{\phi,\alpha}$ across environments, we repeatedly sample 20 random points from

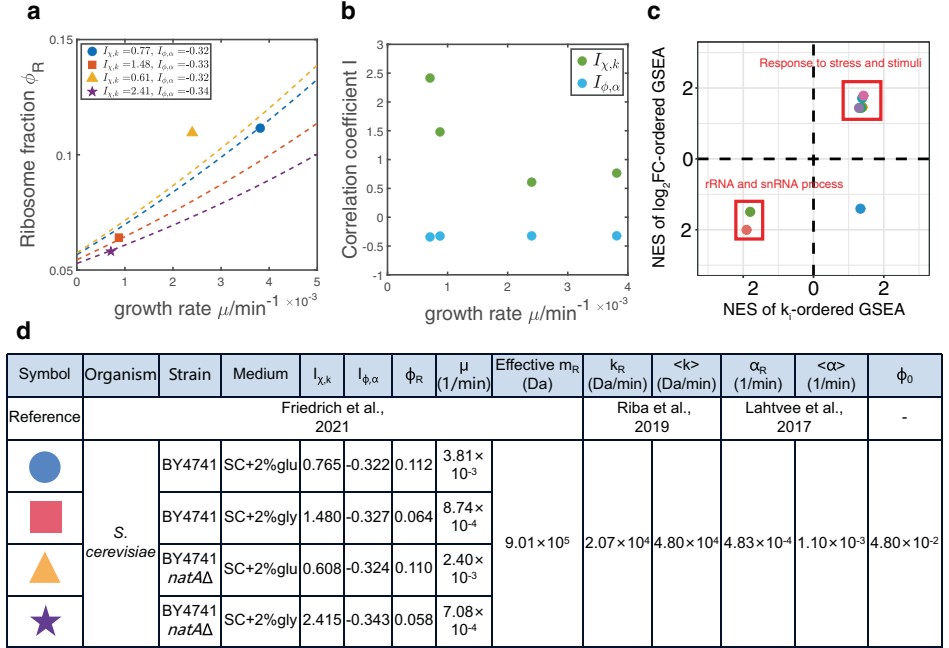

**Fig. 4 Experimental analysis and tests of theoretical predictions. a** Experimental measured $\phi_R$ of *S. cerevisiae* along with the predictions (dashed lines) of our model. **b** The growth rate dependences of the correlation coefficients $I_{\chi,k}$ and $I_{\phi,\alpha}$. **c** The normalized enrichment scores (NES) of GSEA of enriched gene sets. A positive NES of $k_i$-ordered GSEA means that the genes in the corresponding gene set are enriched in the regime of higher $k_i$. A positive NES of $\log_2$FC-ordered GSEA means that the genes in the corresponding gene set are enriched in the regime of increasing $\chi_i$ when the nutrient changes from glucose to glycerol. **d** Summary of the variables and parameters in the analysis of experimental data. The effective mass of ribosomal proteins $m_R$ is calculated based on molecular weights of ribosomal proteins detected in the proteome (Methods). SC synthetic complete medium, Glu glucose, Gly glycerol.

Fig. 3c, f and fit them using Eq. (6) (Methods), imitating the sampling processes in real experiments. We find that when the chosen environments have significantly different $I_{\chi,k}$ and $I_{\phi,\alpha}$, the median root mean squared error RMSE $= 1.69 \times 10^{-2}$ (Fig. 3d). In contrast, in the case of similar environments, RMSE $= 4.44 \times 10^{-3}$ (Fig. 3g). The above results suggest that we can use the fitting error as a criterion of the universality of the growth law curve, which we apply to the experimental data later.

**Experimental tests of theories**. In this section, we test our model using published datasets of *S. cerevisiae*[14] (Methods). For each strain and nutrient quality, we computed the correlation coefficients between the translation speeds and ribosome allocations $I_{\chi,k}$, and the correlation coefficients between the protein degradation rates and protein mass fractions $I_{\phi,\alpha}$. Given the values of $\mu$, $I_{\chi,k}$, and $I_{\phi,\alpha}$, we predicted the fraction of ribosomal proteins $\phi_R$ using Eq. (6) (Fig. 4a, d). We note that one parameter $\phi_0$ is not known experimentally. By choosing a common $\phi_0 = 0.048$, our model predictions nicely match the experimentally measured values of $\phi_R$ (with one data point slightly above the theoretical prediction). We find that regardless of the data processing procedures, the relative relationships between the predicted curves always agree with those of the experimental values (Methods and Fig. S2).

Our model is simplified as we assume that the translation speeds and protein degradation rates do not depend on environments. Remarkably, our model predictions still quantitatively match the experimental observations, suggesting that our assumptions may be reasonable for most situations. While our model cannot predict the growth rate dependence of $\phi_0$, our results show that a constant $\phi_0$ is consistent with three of four data points in Fig. 4a, and the outlier may have a higher $\phi_0$ in that

particular environment. Our analysis cannot exclude the possibility that $\phi_0$ is also environment-specific.

Interestingly, we found that $I_{\phi,\alpha} \approx -0.33$ for all the conditions we computed. However, $I_{\chi,k}$ are negatively correlated with the growth rates, suggesting cells tend to allocate more ribosomes to translate mRNAs with higher $k_i$ in poor nutrient conditions (Fig. 4b). To find out what genes acquire more resources when the environment is shifted, we perform Gene Set Enrichment Analysis (GSEA)[15,16] for wild type cells (Methods) and find that eight gene sets from the Gene Ontology (GO)[17,18] database are enriched in both the GSEA where genes are ordered by $k_i$ and the GSEA where genes are ordered by $\log_2$ fold change ($\log_2$FC) of $\chi_i$ when the nutrient changes (Fig. S3a).

We find that five gene sets related to stress response are enriched in the regime of higher $k_i$ and increasing $\chi_i$ when the environment is changed from 2% glucose to 2% glycerol (Fig. 4c). This is consistent with the environmental stress response (ESR) of *S. cerevisiae* as an adaptation to the shifts of environments[19]. We propose that higher translation speeds of stress response proteins enable cells to respond rapidly to environmental changes, which is evolutionarily advantageous. We also find two gene sets related to the rRNA process enriched in the regime of lower $k_i$ and decreasing $\chi_i$ (Fig. 4c). We also perform GSEA for *natAΔ* cells and get similar results (Fig. S3b, c).

**Applications of theories to experimental growth law curves**. An important application of our theories is that one can estimate the translation speeds by fitting the experimental growth law curve to our model prediction Eq. (6) (Methods). Because there are 6 unknown parameters in the definition of $c_1$, $c_2$, and $c_3$ (Eqs. (23)–(25)), we can estimate three of the parameters given the values of the other three. For the *S. cerevisiae* data from Ref. [6],

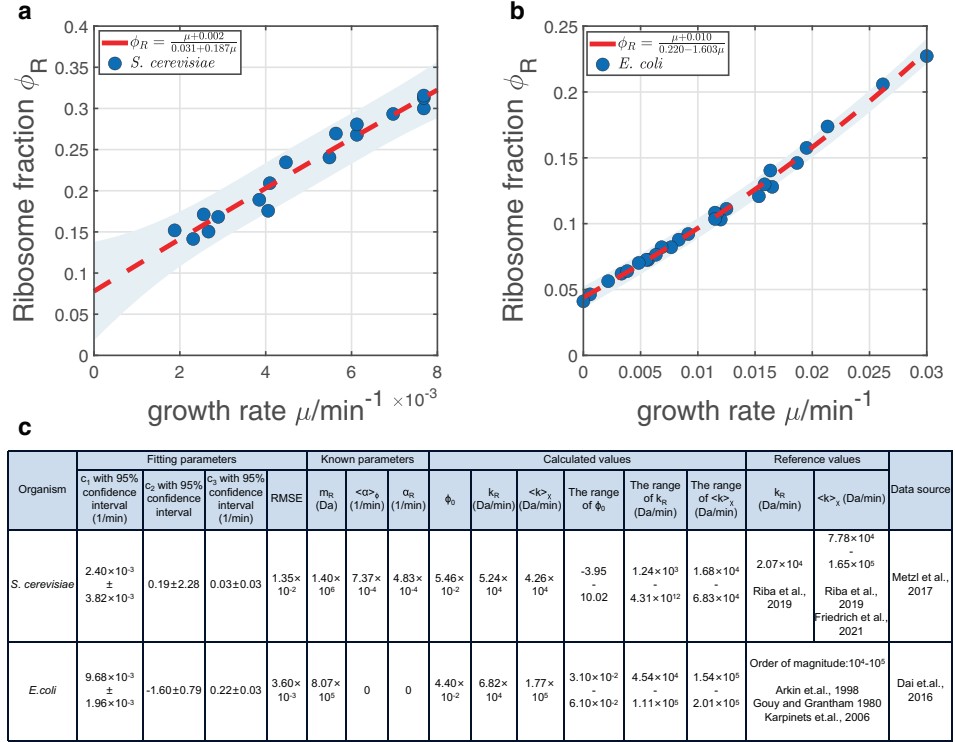

| | Fitting parameters | | | | Known parameters | | | Calculated values | | | | | | Reference values | | Data source |
|---|---|---|---|---|---|---|---|---|---|---|---|---|---|---|---|---|
| Organism | $c_1$ with 95% confidence interval (1/min) | $c_2$ with 95% confidence interval | $c_3$ with 95% confidence interval (1/min) | RMSE | $m_R$ (Da) | $\langle\alpha\rangle_\phi$ (1/min) | $\alpha_R$ (1/min) | $\phi_0$ | $k_R$ (Da/min) | $\langle k\rangle_\chi$ (Da/min) | The range of $\phi_0$ | The range of $k_R$ (Da/min) | The range of $\langle k\rangle_\chi$ (Da/min) | $k_R$ (Da/min) | $\langle k\rangle_\chi$ (Da/min) | |
| S. cerevisiae | $2.40\times10^{-3}$ ± $3.82\times10^{-3}$ | $0.19\pm2.28$ | $0.03\pm0.03$ | $1.35\times10^{-2}$ | $1.40\times10^{6}$ | $7.37\times10^{-4}$ | $4.83\times10^{-4}$ | $5.46\times10^{-2}$ | $5.24\times10^{4}$ | $4.26\times10^{4}$ | $-3.95$ - $10.02$ | $1.24\times10^{3}$ - $4.31\times10^{12}$ | $1.68\times10^{4}$ - $6.83\times10^{4}$ | $2.07\times10^{4}$ Riba et al., 2019 | $7.78\times10^{4}$ - $1.65\times10^{5}$ Riba et al., 2019 Friedrich et al., 2021 | Metzl et al., 2017 |
| E. coli | $9.68\times10^{-3}$ ± $1.96\times10^{-3}$ | $-1.60\pm0.79$ | $0.22\pm0.03$ | $3.60\times10^{-3}$ | $8.07\times10^{5}$ | $0$ | $0$ | $4.40\times10^{-2}$ | $6.82\times10^{4}$ | $1.77\times10^{5}$ | $3.10\times10^{-2}$ - $6.10\times10^{-2}$ | $4.54\times10^{4}$ - $1.11\times10^{5}$ | $1.54\times10^{5}$ - $2.01\times10^{5}$ | Order of magnitude:$10^{4}$-$10^{5}$ Arkin et.al., 1998 Gouy and Grantham 1980 Karpinets et.al., 2006 | | Dai et.al., 2016 |

**Fig. 5 Fitting of the full model to different datasets. a** The non-linear fitting to data of *S. cerevisiae* from Ref. [6]. The shadow represents the 95% prediction interval. **b** The non-linear fitting to data of E. coli from Ref. [7]. **c** Detailed fitting results of (**a**) and (**b**). Note that the reference value of $\langle k\rangle_\chi$ of (**a**) is approximated by $\langle k\rangle(1+I_{\chi,k})$ where the minimal $I_{\chi,k}=0.608$ and the maximal $I_{\chi,k}=2.415$ (Fig. 4d).

we use the experimentally measured degradation rate of ribosomal proteins $\alpha_R$ and the mass of ribosomal proteins $m_R$ as given. We approximate the $\phi$-averaged degradation rate $\langle\alpha\rangle_\phi$ by $\langle\alpha\rangle$ $(1+I_{\phi,\alpha})$ where $I_{\phi,\alpha}=-0.33$, justified by the observations that $I_{\phi,\alpha}$ is independent of environments (Fig. 4b). We find that the fitted parameters $c_1$, $c_2$ and $c_3$ having a wide range of 95% confidence intervals (Fig. 5a, c) with RMSE $=1.35\times10^{-2}$. The inferred ranges of $\phi_0$, $k_R$ and $\langle k\rangle_\chi$ are also unreasonably large (Fig. 5c). All these results suggest that the growth law curves of *S. cerevisiae* are non-universal and large variations of $I_{\chi,k}$ and $I_{\phi,\alpha}$ exist among environments (Fig. 3d).

We also apply our theories to *E. coli*[7] (Fig. 5b). Because most proteins are non-degradable in bacteria[20,21], we set $\alpha_R$ and $\langle\alpha\rangle_\phi$ as 0, and the mass of ribosomal protein $m_R=8.07\times10^{5}Da$[12]. In this case, the fitted parameters have much smaller range of 95% confidence intervals with RMSE $=3.60\times10^{-3}$. The estimated $k_R$, and $\langle k\rangle_\chi$ are consistent with previous studies[22–24] (Fig. 5c). Our analysis of experimental data demonstrates that the translation speed of ribosomal proteins is indeed smaller than the $\chi-$ averaged translation speed, in agreement with experimental observations[9]. Our results suggest that E. coli has similar values of $I_{\chi,k}$ and $I_{\phi,\alpha}$ in the chosen environments of Ref. [7] so that it has a universal growth law curve. In contrast, *S. cerevisiae* appears to have significantly different $I_{\chi,k}$ and $I_{\phi,\alpha}$ across different environments of Ref. [6] so that the growth law curve depends on the chosen environments and therefore non-universal.

**Discussion**. It has been known since the 1950s that the chemical compositions and cell size of bacteria are functions of growth rate and seem to be independent of the medium used to achieve the growth rate[25]. This view has been broadly accepted in the study of bacteria physiologies in the past decades. The growth law

acquired its name because of the independence of the environment. However, recent findings hint at an unforeseen complexity in the growth law. For example, bacterial cell sizes have been shown to depend on the presence of antibiotics, and over-expression of useless proteins[26], and dramatically different cell sizes can exist at the same growth rate[27]. Our study focuses on the growth law regarding the fraction of ribosomal proteins in the proteome and further uncovers the importance of environment-specificity to microbial physiologies. We go beyond the simple translation model and take account of the heterogeneous translation speeds and finite protein degradations. Given the translation speeds and protein degradation rates, our model is completely general and virtually applies to any cells, including both proliferating cells ($\mu > 0$) and non-proliferating cells ($\mu = 0$). In this work, we mainly consider the scenario in which the growth rate changes due to the nutrient quality and the fraction of ribosomal proteins ($\phi_R$) increases monotonically as the growth rate increases.

We demonstrate that the growth law curve generally has the form of Eq. (6). The actual shape of the growth law curve depends on two correlation coefficients: one is between the ribosome allocations and the translation speeds ($I_{\chi,k}$); the other is between the protein mass fractions and protein degradation rates ($I_{\phi,\alpha}$). By analyzing the dataset from[14], we found that $I_{\phi,\alpha}$ is independent of growth rate, while $I_{\chi,k}$ appears to be negatively correlated with the growth rate. This means that cells tend to produce proteins with faster translation speeds in slow-growth conditions, which can be an economic strategy under evolutionary selection. Remarkably, our theoretical predictions of $\phi_R$ reasonably match the experimentally measured values[14]. We note that the upward bending of the growth law curves of bacteria compared with a linear relation appears to hint at an increasing fraction of inactive ribosomes in slow-growth

conditions. While this mechanism appears plausible, it has not been confirmed experimentally as we realize. In our work, we demonstrate that the apparent upward bending can be merely a consequence of heterogeneous translation speeds among proteins and therefore raises caution on the biological interpretation of the shape of the growth law curve.

We apply our model predictions to the growth law curves of *S. cerevisiae*[6] and *E. coli*[7]. In the former case, data fitting to our model prediction is subject to very large uncertainty. This observation agrees with the computed $I_{\chi,k}$ that are variable across conditions using the ribosome profiling and mass spectrometry data from[14] (Fig. 4b). In contrast, the fitting of *E. coli* data exhibits a much smaller uncertainty, suggesting that common $I_{\chi,k}$ and $I_{\phi,\alpha}$ may apply to all the nutrient qualities used in the experiments of Ref. [7]. We expect that this idea can be tested when genome-wide measurements, such as the translation speeds of *E. coli* are available in the future so that critical parameters such as $I_{\chi,k}$ can be calculated for *E. coli*.

We remark that in the absence of heterogeneous translation speeds and protein degradation, the mass fraction of protein $i$, $\phi_i$ must equal the ribosome allocation $\chi_i$. Indeed, these two datasets are often highly correlated among proteins in *E. coli*[12,28]. However, in a more realistic scenario, $\phi_i$ also depends on the translation speed and protein degradation rate. Given the same $\chi_i$, proteins with higher translation speeds or lower degradation rates should have higher mass fractions (Methods). We note that using the current genome-wide datasets of *S. cerevisiae*, the predicted protein mass fractions $\phi_{i,pre}$ based on the ribosome allocations $\chi_i$[14], the translation speeds $k_i$[9], and the protein degradation rates $\alpha_i$[11] do not correlate strong enough with the measured $\phi_i$. We note that these datasets are from different references, and the deviation is likely due to the noise in the measurements of $k_i$ (Table S2). We expect our theories to be further verified when more accurate measurements of translation speeds are available.

For simplicity, in this work, we assume that the translation speeds and protein degradation rates are invariant as the nutrient quality changes. Therefore, we can use the two correlation coefficients $I_{\chi,k}$ and $I_{\phi,\alpha}$ to characterize a particular environment. We remark that our model can be generalized to more complex scenarios in which the translation speeds or protein degradation rates depend on the growth rate[7]. In this case, one just needs to include four additional environment-specific parameters: $k_R$, $\langle k \rangle$, $\alpha_R$, and $\langle \alpha \rangle$.

## Methods

### Derivations of Equation (2). All variables are summarized in Table S3.

Because the total protein mass $M = \sum_i M_i$, we sum over all proteins on both sides of Eq. (1) and obtain

$$\frac{dM}{dt} = \sum_i \frac{dM_i}{dt} = (R - R_0)\sum_i k_i\chi_i - \sum_i \alpha_i M_i. \tag{7}$$

We then divide both sides by $M$ and obtain the expression of the growth rate

$$\mu = \frac{dM/dt}{M} = \frac{R - R_0}{M}\sum_i k_i\chi_i - \frac{\sum_i \alpha_i M_i}{M}$$
$$= \frac{\phi_R - \phi_0}{m_R}\sum_i k_i\chi_i - \sum_i \alpha_i\phi_i, \tag{8}$$

from which Eq. (2) is obtained.

We can also find the changing rate of $\phi_i = M_i/M$ using Eq. (1),

$$\frac{d\phi_i}{dt} = \frac{k_i\chi_i(\phi_R - \phi_0)}{m_R} - \alpha_i\phi_i - \mu\phi_i = 0, \tag{9}$$

which leads to the expression of $\phi_i$ in the steady state as

$$\phi_i = \frac{k_i\chi_i(\phi_R - \phi_0)}{m_R(\mu + \alpha_i)}. \tag{10}$$

Since all proteins grow in the same rate in the steady-state, the growth rates of

protein $i$ defined as

$$\mu_i = \dot{M}_i/M_i = k_i\chi_i(\phi_R - \phi_0)/(m_R\phi_i) - \alpha_i, \tag{11}$$

must be equal to $\mu$, which can be easily verified using Eq. (10). Using the normalization condition $\sum_i \phi_i = 1$, we can write $\phi_i$ using Eq. (10) as

$$\phi_i = \frac{k_i\chi_i/(\mu + \alpha_i)}{\sum_j k_j\chi_j/(\mu + \alpha_j)}. \tag{12}$$

We can also write the normalization condition as

$$1 = \frac{\phi_R - \phi_0}{m_R}\sum_i \frac{k_i\chi_i}{(\mu + \alpha_i)}. \tag{13}$$

### Details of the simplified model without protein degradation. In deriving Eq. (3), we neglect protein degradation and rewrite Eq. (2) as

$$\phi_R = \frac{m_R\mu}{k_R\chi_R + (1 - \chi_R)\langle k \rangle_\chi} + \phi_0. \tag{14}$$

Meanwhile, we compute the growth rate using the auto-catalytic nature of ribosomal proteins,

$$\mu = \frac{\frac{dM_R}{dt}}{M_R} = \frac{k_R\chi_R}{m_R}\left(1 - \frac{\phi_0}{\phi_R}\right). \tag{15}$$

The above equation allows us to replace $\chi_R$ by $\mu$ in Eq. (14), from which we obtain Eq. (3) where

$$a = \frac{k_R - \langle k \rangle_\chi}{k_R(1 - \phi_0) + \langle k \rangle_\chi\phi_0}, \tag{16}$$

$$b = \frac{k_R\langle k \rangle_\chi}{m_R[k_R(1 - \phi_0) + \langle k \rangle_\chi\phi_0]}. \tag{17}$$

### Details of the simplified model with finite protein degradation rates. We now discuss the effects of finite protein degradation rates and assume that the translation speeds are homogeneous and equal to $k$ for all proteins. We rewrite the $\sum_i \alpha_i\phi_i$ term in Eq. (2) such that $\sum_i \alpha_i\phi_i = \alpha_R\phi_R + (1 - \phi_R)\sum_{i=2}^N \alpha_i\widetilde{\phi}_i$. Here, $\phi_i = (1 - \phi_R)\widetilde{\phi}_i$ so that $\sum_{i=2}^N \widetilde{\phi}_i = 1$. We define the $\phi$ – averaged degradation rates over all non-ribosomal proteins as $\langle \alpha \rangle_\phi = \sum_{i=2}^N \alpha_i\widetilde{\phi}_i$. Therefore, Eq. (2) can be written as

$$\phi_R = \frac{\mu + c}{k/m_R + d} + \phi_0. \tag{18}$$

where

$$c = \langle \alpha \rangle_\phi(1 - \phi_0) + \alpha_R\phi_0, \tag{19}$$

$$d = \langle \alpha \rangle_\phi - \alpha_R. \tag{20}$$

To find the sign of $d$, we further rewrite $\langle \alpha \rangle_\phi$ as $\langle \alpha \rangle_\phi = \langle \alpha \rangle(1 + I_{\phi,\alpha})$ where $\langle \alpha \rangle$ is the arithmetic average of degradation rates over all non-ribosomal proteins.

Imagine that we grow cells in multiple environments with equal $I_{\phi,\alpha}$. We assume that the degradation rate of ribosomal protein $\alpha_R$ is slower than the average of non-ribosomal proteins $\langle \alpha \rangle$, which is biologically reasonable since ribosomal proteins are generally non-degraded. Therefore, as long as $I_{\phi,\alpha}$ is not too close to $-1$, which we confirm using experimental data, $d$ is positive since $\alpha_R$ is always smaller than $\langle \alpha \rangle_\phi$. Therefore, our model predicts that the growth law curve is linear given a constant $I_{\phi,\alpha}$ and finite protein degradation decreases the slope relative to the STM. The intercept at $\mu = 0$ is also larger than $\phi_0$. Therefore, a finite fraction of ribosomes are still translating at zero growth rate. We verify the above theoretical predictions by numerical simulations and randomly sample the protein degradation rates that are fixed for all environments, with $\alpha_R < \langle \alpha \rangle$ satisfied.

### Derivations of the full model. In this section we derive the full model considering both the heterogeneities in the translation speeds and protein degradation rates. We rewrite Eq. (2) in the main text as

$$\phi_R = \frac{m_R[\mu + \alpha_R\phi_R + (1 - \phi_R)\langle \alpha \rangle_\phi]}{k_R\chi_R + (1 - \chi_R)\langle k \rangle_\chi} + \phi_0. \tag{21}$$

Meanwhile, the growth rate is

$$\mu = \frac{k_R\chi_R}{m_R}\left(1 - \frac{\phi_0}{\phi_R}\right) - \alpha_R. \tag{22}$$

Combining Eq. (21) and Eq. (22) allows us to solve $\phi_R$ as a function of $\mu$ and we obtain Eq. (6)

$$\phi_R = \frac{\mu + c_1}{c_2\mu + c_3}, \tag{6}$$

where

$$c_1 = \frac{\langle k \rangle_\chi \phi_0}{m_R} + \langle \alpha \rangle_\phi, \tag{23}$$

$$c_2 = 1 - \frac{\langle k \rangle_\chi}{k_R}, \tag{24}$$

$$c_3 = \langle \alpha \rangle_\phi - \frac{\alpha_R \langle k \rangle_\chi}{k_R} + \frac{\langle k \rangle_\chi}{m_R}. \tag{25}$$

It is straightforward to find that the condition for Eq. (6) to be monotonically increasing is that $c_3 > c_1 c_2$. Using the above expressions, we find that

$$c_3 - c_1 c_2 = \frac{\langle k \rangle_\chi (1 - \phi_0)}{m_R} + \frac{\langle k \rangle_\chi^2 \phi_0}{k_R m_R} + \frac{\langle k \rangle_\chi (\langle \alpha \rangle_\phi - \alpha_R)}{k_R}. \tag{26}$$

We find that the first two terms are always positive, and the last term is positive as long as $I_{\alpha,\phi}$ is not too close to $-1$. Therefore, the $\phi_R(\mu)$ curve must increase monotonically. It is straightforward to find that the second derivative of the $\phi_R(\mu)$ curve is proportional to $(c_1 c_2 - c_3) c_2$, which is always positive as long as $I_{\chi,k}$ is not too close to $-1$.

**Details of the numerical simulations.** We summarize the parameters we use in the numerical simulations in Table S1. We consider a cell with 4000 genes. We set the elongation speed $k_i$ and the degradation rates $\alpha_i$ of non-ribosomal genes to follow lognormal distributions. We set $k_R = 2.07 \times 10^4$ Da/min, $\langle k \rangle = 4.80 \times 10^4$ Da/min, $\alpha_R = 4.83 \times 10^{-4}$ min$^{-1}$, and $\langle \alpha \rangle = 1.10 \times 10^{-3}$ min$^{-1}$ as the experimentally measured values of *S. cerevisiae*[9,11]. The coefficients of variation (CV) of the lognormal distributions can be found in Table S1. In all simulations, we set $\phi_0 = 0.08$. We note that in Fig. 2a, we set $\alpha_i = 0$ for all proteins and in Fig. 2b, we set $k_i = \langle k \rangle$ for all proteins.

To simulate a random environment, we generate a random $\chi_R$. Meanwhile, a lognormal distribution of $\chi_i$ of non-ribosomal genes is also randomly generated. The CV of the lognormal distribution is included in Table S1. We then search for the $\phi_R$ and $\mu$ that simultaneously satisfy Eq. (22) and Eq. (13). $\phi_i$, $I_{\chi,k}$ and $I_{\phi,\alpha}$ are then calculated using Eq. (10), Eq. (4) and Eq. (5), respectively. For a chosen pair of $I_{\chi,k}$ and $I_{\phi,\alpha}$, the predicted $\phi_R(\mu)$ curve is obtained using Eq. (6). To obtain Fig. 3d, g, we randomly sample 20 points from Fig. 3c, f respectively, fit them using Eq. (6), and calculate the resulting RMSE. We repeat the above process 5000 times.

**Details of the experimental data analysis.** For the ribosome profiling data[14], we first trim the adapter with Cutadapt (version 3.4)[29]. Then we use Bowtie2 (version 2.4.2)[30] to eliminate ribosomal RNAs (rRNA) as mentioned in[31]. The cleaned reads are then mapped to *S. cerevisiae* genome R64.1.1 with HISAT2 (version 2.2.1)[32]. Read counts are then generated with featureCount (version 2.0.1)[33]. We then manually eliminate the non-coding RNAs. The ribosome allocation $\chi_i$ is calculated based on the mean count fraction of all samples (Supplementary Data 3).

For the proteomics data[14], we perform the absolute quantification (or the in-sample relative quantification) of proteins based on the intensities of peptides using xTop (version 1.2)[12]. The intensity ratio of 2 proteins in the same sample of proteomics data does not directly represent the real abundance (either the mass or the copy number) ratio, so the abundance fraction can not be replaced with the intensity fraction[12,34]. xTop is a software that accurately calculates the in-sample relative protein copy number with the maximum a posteriori probability (MAP) algorithm[12]. We then calculate all proteins' mass fraction $\phi_i$ with the xTop results and the protein molecular mass (Supplementary Data 2). In[12], the authors further calibrated $\phi_i$ with ribosome profiling data assuming homogeneous $k_i$. In this work, we alternatively calibrate $\phi_i$ with $L^{-0.57}$ where $L$ is the protein length, as mentioned in[12]. Calibration with $L^{-0.57}$ is independent of ribosome profiling data, although it reduces the distance between $\chi_i$ and calibrated $\phi_i$[12]. We also show the result with calibration of $L^{-1}$ or without calibration in Fig. S2b, c. To compute $\phi_R$, we sum up the $\phi_i$ of all proteins annotated as the cytoplasmic ribosomal protein in the Saccharomyces Genome Database (SGD).

For the elongation speed $k_i$, we first calculate $v_i$ as mentioned in[9]. $k_i$ is then calculated using the relationship $k_i = v_i a_i$ where $v_i$ is the number of translated amino acids per unit time, and $a_i$ is the averaged mass of amino acids over the sequence of protein $i$ (Supplementary Data 1). For the degradation rate $\alpha_i$, data is obtained from[11]. We calculate the experimental $I_{\chi,k}$, $I_{\phi,\alpha}$, $\langle k \rangle$ and $\langle \alpha \rangle$ for non-ribosomal genes that exist in all data sets of $\chi_i$, $\phi_i$, $k_i$ and $\alpha_i$. We also calculate the $\chi$-averaged $k$ of ribosomal proteins as $k_R$ and $\phi$-averaged $\alpha$ of ribosomal proteins as $\alpha_R$.

For the molecular mass of the ribosome, we calculate the effective $m_R$. Considering the efficiency of the mass spectrometry (MS), not all proteins can be detected. Therefore, we define the effective $m_R$ as the molecular weights of ribosomal proteins detected in the proteome. Because most of the ribosomal proteins can be expressed by two paralogous genes in *S. cerevisiae*, we count the average molecular mass when both proteins of the paralogs are detected in the proteome. We also show our predictions of $\phi_R$ using the actual ribosome mass ($m_R = 1.40 \times 10^6$ Da) in Fig. S2a.

For the growth rate $\mu$ (Supplementary Data 4), it is obtained from the growth curve, $OD_{600}$ versus time from[14] with the method mentioned in[35]. Briefly, the slopes of $\ln(OD_{600})$ versus time in 5-point windows are calculated. Then windows with slopes at least 95% of the maximum slope are extracted. The slope of points within these windows is calculated as the growth rate. With these results, we predict the corresponding $\phi_R(\mu)$ curves and compare them with the experimental data points.

We further calculate the predicted mass fraction $\phi_{i,pre}$ of non-ribosomal proteins with Eq. (12). Pearson correlation coefficients $\rho$ between $\phi_{i,pre}$ and $\phi_i$ are calculated. We also compute $\rho$ under the assumptions that $\alpha_i = 0$ or $k_i = \langle k \rangle$ (Table S2).

For GSEA analysis, we first perform the differential expression analysis on the ribosome profiling data of WT or *natA*Δ cells using the package DEseq2 (version 1.24.0)[36] in R (version 3.6.1). The $\log_2$ fold changes in counts when cells changed from SC+2% glucose to SC+2% glycerol, the p-value of the two-sided Wald test, and the FDR q-values are calculated. Ribosomal genes and genes with FDR q-value > 0.05 are eliminated. We then pick out genes that also exist in the data sets of $k_i$. GSEA on these genes is performed twice using the R package clusterProfiler (version 3.12.0)[37] and org.Sc.sgd.db (version 3.8.2)[38]. In the first GSEA, genes are ordered by the $\log_2$ fold change (denoted as $\log_2$FC-ordered GSEA). In the second GSEA, genes are ordered by $k_i$ (denoted as $k_i$-ordered GSEA). We then find the common gene sets from GO database[17,18] enriched in these two GSEA. The cut-off criteria are set as the p-value < 0.05 (single-sided permutation test) and the FDR q-value < 0.25. The number of permutations used in the analysis is $10^5$.

**Details of fitting in Fig. 5.** Nonlinear fitting is performed with MATLAB (version R2020b). We obtain the fitting parameters $c_1$, $c_2$ and $c_3$ with their 95% confidence intervals, and then compute $\phi_0$, $k_R$ and $\langle k \rangle_\chi$ using Eqs. (23), (24), (25). To compute the ranges of these values, we numerically find the maximum and the minimum value of the multivariate functions $\phi_0(c_1, c_2, c_3)$, $k_R(c_1, c_2, c_3)$ and $\langle k \rangle_\chi(c_1, c_2, c_3)$ as their upper and lower bounds, where the ranges of $c_1$, $c_2$ and $c_3$ are their 95% confidence intervals.

**Statistics and reproducibility.** We use the two-sided Wald test in the differential expression analysis of the ribosome profiling data. In GSEA, we use a single-sided permutation test. As for reproducibility, no biological experiments are performed in our work, and all data are acquired from public repositories (see Data Availability).

**Reporting summary.** Further information on research design is available in the Nature Research Reporting Summary linked to this article.

## Data availability

The Ribosome profiling data from[14] was deposited at GEO (GSE140255) (link https://www.ncbi.nlm.nih.gov/geo/query/acc.cgi?acc=GSE140255). The Proteomics data from[14] was deposited at ProteomeXchange (PXD015217) (link http://proteomecentral.proteomexchange.org/cgi/GetDataset?ID=PXD015217). The growth rate data was acquired from the figures of OD600-versus-time curves in[14]. The Ribosomal protein list was acquired from Saccharomyces Genome Database (SGD) (link https://yeastgenome.org). The data needed calculating elongation speeds was acquired from the supplementary materials of[9]. The protein degradation rate data was acquired from the supplementary materials of[11]. The $\phi_R$-$\mu$ data of *E. coli* was acquired from the supplementary materials of[7]. The $\phi_R$-$\mu$ data of *S. cerevisiae* was acquired from the figure in[6]. Calculated data of $k_i$, $\phi_i$, $\chi_i$, and $\mu$ have been provided in Supplementary Data 1-4.

## Code availability

All codes for mathematical simulations and data analysis are available in the following link (https://github.com/QirunWang/Codes-for-Environment-specificity-and-universality-of-the-growth-law).

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

## Acknowledgements

We thank Poyi Ho for useful discussions related to this work. The research was funded by National Key R&D Program of China (2021YFF1200500) and supported by grants from Peking-Tsinghua Center for Life Sciences.

## Author contributions

Q.W. and J.L. conceived, designed, and carried out the theoretical and numerical part of this work. Q.W. performed the analysis of experimental data. All the authors contributed to the preparation of the manuscript.

## Competing interests

The authors declare no competing interests.
