## [Peer Review File · Communications Biology]

Reviewers' comments:

Reviewer #1 (Remarks to the Author):

The main contribution of Qirun Wang and Jie Lin's manuscript is to extend the linear relation between the fraction of ribosomal proteins in the proteome and the growth rate of proliferating cells to a nonlinear relation in the case of heterogeneous translation speeds and protein degradation. They showed that this relation curves are generally environment-specific, depending on two kinds of correlations: one is between the ribosome allocations and the translation speeds and the other is between the protein mass fractions and protein degradation rates. They also verified their predictions by analyzing data of two real biological examples: *Saccharomyces cerevisiae* and *Escherichia coli*. I think that this manuscript contains interesting ingredients, but I have the following comments.

- (1) I suggest that the title is revised as "the relation between the fraction of ribosomal proteins in the proteome and the growth rate of proliferating cells is generally environment-specific", at least the word 'environmental' in the title should be revised as 'environment'.
- (2) The logic for deriving Eq. (2) from Eq. (1) seems unclear. I notice that the authors first presented the relation of the mass fraction of protein i with the growth rate [i.e., Eq. (2)], and then gave interpretations directly below Eq. (2), including the derivation of Eq. (3). I think that this arrangement would be difficult for readers to understand, at least for me.
- (3) A more important point is that the uncertainty of the environment would lead to the random growth of protein mass and the random allocation of ribosomes. It seems to us that the authors did not consider this point. If deriving analytical results is difficult when this stochasticity is considered, I suggest that the authors give explanations (however, I notice that the authors considered stochastic factors in their numerical simulations).
- (4) On the whole, the qualitative relation between the fraction of ribosomal proteins in the proteome and the growth rate of the total protein mass in the case of considering heterogeneous translation speeds and protein degradation is fundamentally similar to that in the case of not considering heterogeneous translation speeds and protein degradation, at least without essential difference between them. This can be seen from the figures demonstrated in the manuscript. Therefore, I have a little bit concern about the biological significance of this study although the nonlinear relation (i.e., Eq. (7)) derived by the authors is impressive.

Reviewer #2 (Remarks to the Author):

The authors propose a model of protein synthesis for cell growths, which takes into account the heterogeneities of the protein translation and degradation rates. They theoretically prove that the growth law connecting the fractions of ribosomal proteins and the cell growth rates is upward-bending due to the heterogeneity of protein translation rates, particularly the slower translation speed of ribosomal proteins compared to that of non-ribosomal proteins. The model is then used to interpret prior experimental data of *S. cerevisiae* and *E. coli*. It is shown that the growth laws are generally environmental-specific or non-universal, meaning that the coefficients in the theoretically derived growth law equation (Eq. 7) are varied with nutrient conditions, as shown for the case of *S. cerevisiae*. The growth law can be also universal, i.e. with small variations of the coefficients, as shown for the case of *E. coli*.

The study is interesting. The proposed model is meaningful, more general and goes beyond the previously used STM model. There are however several issues that need clarification:

- 1) A major claim of this study is that the mass fraction of inactive ribosomes (ϕ_0) is constant for a given organism, i.e. it does not depend on environmental conditions. However, by looking more carefully at the data this claim seems not well supported. This is seen in Fig. 4 with the data for *S.*

cerevisiae: the fit in Fig. 4a is not good for one data point (the triangle) – a good fit would require ϕ_0 to be significantly increased. Also, the positions of the square and the star points in Fig. 4a are not consistent with the data given in Fig. 4d, which shows that they have the same ϕ_R (0.06). I am not sure whether this is just number rounding, but this may lead to a variation in ϕ_0 . Furthermore, the data in Fig. 5c for *S. cerevisiae* also shows that the variation in ϕ_0 is quite large, from -3.92 to 9.93 (?). It is possible that ϕ_0 also depend on the environment like the other parameters.

2) What happens if the data in Fig. 4a are included onto Fig. 5a, as they are both for *S. cerevisiae*? The different units of the growth rates in these two figures make it difficult to judge whether the two datasets fall on the same growth law region.

3) Caption of Fig. 5 says that the range of the correlation coefficient $I_{\chi,k}$ can be found in Figure 4c. Actually, there is no such information in Fig. 4c. One would be also interested in the values of the correlation coefficients for *E. coli*, which are missing.

4) The parameters a and b in Eq. (4) are not shown in the Methods section.

Reviewer #1 (Remarks to the Author):

The main contribution of Qirun Wang and Jie Lin's manuscript is to extend the linear relation between the fraction of ribosomal proteins in the proteome and the growth rate of proliferating cells to a nonlinear relation in the case of heterogeneous translation speeds and protein degradation. They showed that this relation curves are generally environment-specific, depending on two kinds of correlations: one is between the ribosome allocations and the translation speeds and the other is between the protein mass fractions and protein degradation rates. They also verified their predictions by analyzing data of two real biological examples: *Saccharomyces cerevisiae* and *Escherichia coli*. I think that this manuscript contains interesting ingredients, but I have the following comments.

We thank Reviewer 1 for their careful reading and appreciation that our work contains interesting ingredients. We have made important changes in the revised manuscript as suggested by Reviewer 1.

(1) I suggest that the title is revised as "the relation between the fraction of ribosomal proteins in the proteome and the growth rate of proliferating cells is generally environment-specific", at least the word 'environmental' in the title should be revised as 'environment'.

Answer: We apologize for the typo and appreciate the new suggested title. We noticed that our previous title was a bit too long. In the revised manuscript, we have changed the word 'environmental' in the title to 'environment' as suggested and shortened the title to "Environment-specificity and universality of the microbial growth law."

(2) The logic for deriving Eq. (2) from Eq. (1) seems unclear. I notice that the authors first presented the relation of the mass fraction of protein i with the growth rate [i.e., Eq. (2)], and then gave interpretations directly below Eq. (2), including the derivation of Eq. (3). I think that this arrangement would be difficult for readers to understand, at least for me.

Answer: We apologize for the confusion regarding the logic of the derivations. In the revised manuscript, we have simplified the derivation of Eq. (3) (now Eq. (2) in the revised manuscript) without using Eq. (2) in the previous manuscript and leave the derivation details in the first section of Method:

"We define the total protein mass $M = \sum M_i$, and the protein mass fraction $\phi_i = \frac{M_i}{M}$. Using Eq. (1), we find the fraction of ribosomal proteins in the proteome in the steady state, (see detailed derivations in Methods)

$$\phi_R = \frac{m_R(\mu + \sum \alpha_i \phi_i)}{\sum k_i \chi_i} + \phi_0 \quad (2)$$

Here μ is the growth rate of the total protein mass $M = \dot{M}/M$, and m_R is the total amino acid mass of a single ribosome. ϕ_0 is the mass fraction of inactive ribosomes, which we assume to be constant for simplicity. In this work, $i = 1$ is reserved for ribosomal proteins so that $\phi_1 = \phi_R$, $k_1 = k_R$, and $\alpha_1 = \alpha_R$. Here, k_R and α_R are the effective translation speed and degradation rate of the coarse-grained ribosomal protein averaged over all ribosomal proteins. They are approximately independent of environments due to the tight regulation of relative doses of different ribosomal proteins (Li, et al., *Cell*, 2014) and their generally low degradation rates. It is easy to find that if all proteins have the same translation speed ($k_i = k$ for all i) and protein degradations are negligible ($\alpha_i = \alpha_0$), Eq. (2) is reduced to the STM."

Furthermore, we notice that Eq. (2) in the previous manuscript was not used in the main text, and we have removed it from the main text.

(3) A more important point is that the uncertainty of the environment would lead to the random growth of protein mass and the random allocation of ribosomes. It seems to us that the authors did not consider this point. If deriving analytical results is difficult when this stochasticity is considered, I suggest that the authors give explanations (however, I notice that the authors considered stochastic factors in their numerical simulations).

Answer: We appreciate Reviewer 1 for raising this important point. In the revised manuscript, we have included new simulations incorporating the uncertainty of the environment by introducing noises to the elongation speeds k_i and the allocation fraction χ_i . We find that both noises do not affect our conclusions qualitatively, validating our theoretical expressions of the growth law curves given fixed $I_{\chi,k}$ and $I_{\phi,\alpha}$. Adding noises to the elongation speeds and allocation fraction makes the resulting growth law curves even noisier and, therefore, does not affect our main conclusion that the growth law curve is generally environment-specific.

In the revised manuscript, we have clarified this point in the section "Universal and non-universal growth law curves":

"We note that the uncertainty of real environments often leads to random production of proteins and random allocation of ribosomes. To address this question, we also simulate models in which noises exist in the elongation speeds k_i and the allocation fractions χ_i . We find that both noises do not affect our conclusions qualitatively (Supplementary Figure S1). Note that adding noises to the elongation speeds and allocation fractions only makes the resulting growth law curves even noisier and therefore does not affect our main conclusion that the growth law curve is generally environment-specific, as we show later."

We have added a new figure (Figure S1) on the simulations of random elongation speeds and ribosome allocation fractions to the Supplementary Material (see the figure below).

Figure S1: Simulations including random elongation speeds and random allocation fractions of ribosomes. Simulation details are the same as the main text except that we add noises to the elongation speed k_i and the allocation fraction of ribosomes χ_i , respectively. (Left) Simulations including random elongation speeds. We add a normal distributed noise $\xi_k \sim N(0, (\frac{k_i}{10})^2)$ to the elongation speed of each protein. (Right) Simulations including random allocations of ribosomes. We add a normal distributed noise $\xi_\chi \sim N(0, (\frac{\chi_i}{10})^2)$ to the allocation fraction χ_i and then normalize them so that the sum of χ_i equals 1.

(4) On the whole, the qualitative relation between the fraction of ribosomal proteins in the proteome and the growth rate of the total protein mass in the case of considering heterogeneous translation speeds and protein degradation is fundamentally similar to that in the case of not considering heterogeneous translation speeds and protein degradation, at least without essential difference between them. This can be seen from the figures demonstrated in the manuscript. Therefore, I have a little bit concern about the biological significance of this study although the nonlinear relation (i.e., Eq. (7)) derived by the authors is impressive.

Answer: We appreciate Reviewer 1 for pointing out this concern, and we apologize that we did not explain well the biological significance of our study in the previous manuscript. In the revised manuscript, we have clarified better the biological significance of our study in the Discussion section regarding the following two points:

1. It has been known since the 1950s that bacteria's chemical compositions and cell size are functions of growth rate and seem to be independent of the medium used to achieve the growth rate (Schaechter et al., *Microbiology*, 1958). This view has been broadly accepted in the study of bacteria physiologies in the past decades. The "growth law" acquired its name because of the independence of the environment. However, recent findings hint at an unforeseen complexity in the growth law. For example, bacterial cell sizes have been shown to depend on the presence of antibiotics and overexpression of useless proteins (Basan et al., *Molecular Systems Biology*, 2015). Dramatically different cell sizes can exist at the same growth rate (Vadia &

Levin, *Current Opinion in Microbiology*, 2015). Our study focuses on the growth law regarding the fraction of ribosomal proteins in the proteome and further uncovers the importance of environment-specificity to bacterial physiologies.

2. Previous works have speculated that the upward bending of the growth law curve compared with a linear relation in slow-growth conditions is due to the increasing fraction of inactive ribosomes. While this mechanism appears plausible, it has not been confirmed experimentally as we realize. In our work, we demonstrate that the apparent upward bending can be merely a consequence of heterogeneous translation speeds among proteins and therefore raises caution on the biological interpretation of the shape of the growth law curve.

Reviewer #2 (Remarks to the Author):

The authors propose a model of protein synthesis for cell growths, which takes into account the heterogeneities of the protein translation and degradation rates. They theoretically prove that the growth law connecting the fractions of ribosomal proteins and the cell growth rates is upward-bending due to the heterogeneity of protein translation rates, particularly the slower translation speed of ribosomal proteins compared to that of non-ribosomal proteins. The model is then used to interpret prior experimental data of *S. cerevisiae* and *E. coli*. It is shown that the growth laws are generally environmental-specific or non-universal, meaning that the coefficients in the theoretically derived growth law equation (Eq. 7) are varied with nutrient conditions, as shown for the case of *S. cerevisiae*. The growth law can be also universal, i.e. with small variations of the coefficients, as shown for the case of *E. coli*.

The study is interesting. The proposed model is meaningful, more general and goes beyond the previously used STM model. There are however several issues that need clarification:

We thank Reviewer 2 for their appreciation that our work provides a meaningful and more general model. We have made important changes in the revised manuscript as suggested by Reviewer 2.

1) A major claim of this study is that the mass fraction of inactive ribosomes (ϕ_0) is constant for a given organism, i.e. it does not depend on environmental conditions. However, by looking more carefully at the data this claim seems not well supported. This is seen in Fig. 4 with the data for *S. cerevisiae*: the fit in Fig. 4a is not good for one data point (the triangle) – a good fit would require ϕ_0 to be significantly increased. Also, the positions of the square and the star points in Fig. 4a are not consistent with the data given in Fig. 4d, which shows that they have the same ϕ_R (0.06). I am not sure whether this is just number rounding, but this may lead to a variation in ϕ_0 . Furthermore, the data in Fig. 5c for *S. cerevisiae* also shows that the variation in ϕ_0 is quite large, from -3.92 to 9.93 (?). It is possible that ϕ_0 also depend on the environment like the other parameters.

Answer: We appreciate Reviewer 2 for raising this important point. We would like to clarify that our work cannot provide a definite conclusion on the mass fraction of inactive ribosomes. We completely agree with Reviewer 2 that the outlier in Fig. 4a may suggest a higher ϕ_0 in that particular environment.

We do not mean that the mass fraction of inactive ribosomes is universally constant, and we apologize for the confusion. In fact, whether ϕ_0 is constant is still under debate. Dai et al. proposed that the fraction of inactive ribosomes may increase as the growth rate decreases in *E. coli* (Dai et al. *Nature Microbiology* 2016). Meanwhile, Metzl et al. found a constant $\phi_0 = 8\%$ fits well with their data of *S. cerevisiae* (Metzl et al. *eLife*, 2017). In our study, we also find a constant $\phi_0 = 4.8\%$ can fit 3 out of 4 points of the experimental data.

We would like to clarify that the square and star points in Fig. 4a have different ϕ_R indeed and the inconsistency with the data in Fig. 4d is due to number rounding. This does not affect our conclusion that a constant ϕ_0 can fit both data points.

As for the large ϕ_0 range shown in Fig. 5c, we would like to clarify that this is merely the range inferred from the growth law curve in Fig. 5a, which demonstrates that the growth law curve in Fig. 5a is subject to large variations among environments and therefore non-universal.

We have fixed the number-rounding problem in Fig. 4d in the revised manuscript.

Symbol	Organism	Strain	Medium	$I_{\chi,k}$	$I_{\phi,\alpha}$	ϕ_R	μ (1/min)	Effective m_R (Da)	k_R (Da/min)	$\langle k \rangle$ (Da/min)	α_R (1/min)	$\langle \alpha \rangle$ (1/min)	ϕ_0
Reference	Friedrich et al., 2021							Riba et al., 2019		Lahtvee et al., 2017		-	
	S. cerevisiae	BY4741	SC+2%glu	0.765	-0.322	0.112	3.81×10^{-3}	9.01×10^5	2.07×10^4	4.80×10^4	4.83×10^{-4}	1.10×10^{-3}	4.80×10^{-2}
		BY4741	SC+2%gly	1.480	-0.327	0.064	8.74×10^{-4}						
		BY4741 Δ Naa10	SC+2%glu	0.608	-0.324	0.110	2.40×10^{-3}						
		BY4741 Δ Naa10	SC+2%gly	2.415	-0.343	0.058	7.08×10^{-4}						

We have also added more clarifications on the inferred fraction of inactive ribosomes in the section "Applications of theories to experimental growth law curves":

"The inferred ranges of ϕ_0 , k_R and $\langle k \rangle_\chi$ are also unreasonably large (Figure 5c). All these results suggest that the growth law curves of *S. cerevisiae* are non-universal and large variations of $I_{\chi,k}$ and $I_{\phi,\alpha}$ exist among environments (Figure 3d)."

2) What happens if the data in Fig. 4a are included onto Fig. 5a, as they are both for *S. cerevisiae*? The different units of the growth rates in these two figures make it difficult to judge whether the two datasets fall on the same growth law region.

Answer: We apologize for the confusion of the different units. In the revised manuscript, we have used the same units for the growth rates (min^{-1}) in Fig. 4 and 5.

We also combined the data in Fig. 4a and Fig. 5a into the same figure. We found that they do not fall on the same growth law region, which may be due to batch effects and/or different definitions of ribosome fractions in different experiments (see the figure below).

3) Caption of Fig. 5 says that the range of the correlation coefficient $I_{\chi,k}$ can be found in Figure 4c. Actually, there is no such information in Fig. 4c. One would be also interested in the values of the correlation coefficients for *E. coli*, which are missing.

Answer: We apologize for the typo; it should be Figure 4d. In the revised manuscript, we have fixed it and the range of $I_{\chi,k}$ is from 0.608 to 2.415.

Regarding the $I_{\chi,k}$ values of *E. coli*, we would like to mention that we are still unable to calculate them due to the lack of data on the elongation speeds. What we can tell now based on the analysis we have is that the $I_{\chi,k}$ values of *E. coli* are less variable across different environments, indicated by the small ranges of the 95% confidence intervals of the fitted parameters, as shown in Fig. 5b, c.

In the revised manuscript, we have modified the caption of Fig. 5:

"Note that the reference value of $\langle k \rangle_{\chi}$ of (a) is approximated by $\langle k \rangle (1 + I_{\chi,k})$ where the minimal $I_{\chi,k} = 0.608$ and the maximal $I_{\chi,k} = 2.415$ (Figure 4d)."

We also clarified why the values of the correlation coefficients for *E. coli* are missing in the Discussion section:

"In contrast, the fitting of *E. coli* data exhibits a much smaller uncertainty, suggesting that common $I_{\chi,k}$ and $I_{\phi,\alpha}$ may apply to all the nutrient qualities used in the experiments of Dai et al., *Nature Microbiology* 2016. We expect that this idea can be tested when genome-wide measurements, such as the translation speeds of *E. coli* are available in the future so that critical parameters such as $I_{\chi,k}$ can be calculated for *E. coli*."

4) The parameters a and b in Eq. (4) are not shown in the Methods section.

Answer: We apologize for this mistake. In the revised manuscript, we have included the expressions of the parameters a and b in the Methods section "Details of the simplified model without protein degradation."

REVIEWERS' COMMENTS:

Reviewer #1 (Remarks to the Author):

After having read the authors' response letter and the revised file, I think that this manuscript was improved, and that in particular, they fixed our concerns raised in the last manuscript. Currently, I am satisfied with the revised version. I have no more comments, and directly recommend the manuscript for acceptance.

Reviewer #2 (Remarks to the Author):

The authors have revised the paper. Their revisions and their responses sufficiently address the concerns in my previous report. I agree with their clarification about ϕ_0 . This is an interesting work. I recommend publication of the paper in Communications Biology.